# Universities That Learn to Tackle the Challenges of Sustainability: Case Study of the University of Córdoba (Spain)

**Antonio Gomera** [1,2,3], **Miguel Antúnez** [2,*] **and Francisco Villamandos** [2,3]

1 Environmental Protection Service (SEPA), Campus Rabanales, University of Córdoba, 14014 Córdoba, Spain; agomera@uco.es
2 Sustainability Office, Campus Rabanales, University of Córdoba, 14014 Córdoba, Spain; bv1vitof@uco.es
3 Research Group 'SEJ-049: Educational Evaluation and Innovation', Faculty of Education, University of Córdoba, 14071 Córdoba, Spain
* Correspondence: m.antunez@uco.es

**Abstract:** On their path to sustainability, universities must consider both individual and organizational components. Universities are organizations that also have the capacity to learn and evolve. By means of an analysis conducted over the past 20 years at the University of Córdoba (Spain), this article identifies the variables present in the University's environmental sustainability process, characterizing its evolution through different stages and proposing an organizational model that orders these variables into a system within the framework of complexity. This model highlights the importance of a scientific-technical structure as catalyst, facilitator, and attractor of transformative flows within the organization, which could be a key component of its evolution towards sustainability. It also underscores the possibility of using environmental awareness and the perceived norm as indicators of the system. This characterization reveals the potential of these variables as indicators of progress and anchoring points for the permanent monitoring of the system, and it will also help to design potentially more effective and forceful actions and could prove valuable as a comparison indicator between universities.

**Keywords:** sustainability; environment; universities; organizational learning; complexity

## 1. Introduction

### 1.1. Sustainability, Complexity and Organizational Learning

Over the past few centuries, human activities have had important and diverse effects on natural systems, effects that are increasingly fast-paced and wide-ranging. We are changing our planet more rapidly than we understand [1]. This phenomenon is also referred to as "global environmental change" [2]. This global change is a social fact, not only because it is largely caused by human activities, but also because its consequences ultimately affect societies and people [3]. In short, the implications of our current relationship with the environment are revealing a systemic crisis [4], a genuine crisis of civilization [5].

The necessary paradigm shift that would bring us closer to finding a solution to this problem points to a transformation in the economic, political, and cultural order. Such a transformation would be inconceivable without the corresponding evolution in human awareness and behavior [6], and in which the educational settings such as universities are key agents. The complex nature of the problem we are facing calls for a new way of thinking, typical of a systemic vision where, as Mora [7] (p. 10) states, "the act of separating, typical of modernity and of techno-scientific hyper-specialization,

is complemented by the act of uniting, combining, connecting, and interacting". A systemic way of thinking that would draw us away from a systemic crisis by seeking vital sustainability [8].

To deal effectively with the profound transformations promoted by this paradigm shift, we must turn to the posits of Chaos Theory [9], transdisciplinarity [10], or the perspective of complexity. We understand that these three contributions constitute different perspectives and with very different methods and assumptions, from which the reality of a complex world is contemplated. A reality in which the complexity approach proposes a more complete one.

The epistemology of complexity reformulates and moves beyond the contributions of other approaches [11], integrating an open and inquiring attitude and rejecting simplification in favor of a new way of contemplating the world based on recurrence and the use of metadisciplinary concepts [12]. The science of complexity studies world phenomena by assuming their complex nature and seeks predictive models that incorporate the existence of chance and uncertainty, as a way of approaching reality that extends not only to experimental sciences but also to social sciences [13].

For some authors [14,15], awareness of uncertainty and complexity along with the contributions of the systemic approach and Chaos Theory have brought about a paradigm shift in the way organizations are understood, moving from a linear vision, where there is a clear cause–effect relationship, to a non-linear one.

Salazar Duque [15] attempts to exemplify the differences involved in approaching management systems in these ways, comparing the traditional approach with the chaos approach (Table 1).

**Table 1.** Comparison between the traditional approach and the chaos approach: Prigogine's Chaos Theory and its usefulness in managing complex systems.

| Traditional Approach | Chaos Approach |
| --- | --- |
| It is possible to predict the behavior of any future state of the system through a simple cause-effect equation. | There is no proportionality in the cause-effect relationship. The future is uncertain, and the system reacts unpredictably; the system does not evolve continuously. |
| The whole is the sum of its parts. | The complex whole is made up of infinite iterations of a simple pattern that is repeated on different scales. |
| Chaos is synonymous with disorder and can be avoided by controlling the system as much as possible. | There is a close relationship between chaos and order, and indeed one leads to the other by means of a dynamic process. It is not about avoiding chaos, but about using it to self-organize your system through an 'attractor'. |
| The system does not change suddenly; if it does, it is due to an error that has not been properly controlled. | A minor disturbance can suddenly trigger explosive changes within the system. |
| An item cannot belong to a set and its complement at the same time. | The relationship between items and sets is not just yes or no; it is a matter of more or less. |

Source: Salazar Duque [15].

In this new paradigmatic approach, small actions can have unpredictable consequences, implying a different approach to organizational management. This approach brings different innovations, including the promotion of aspects that facilitate so-called 'organizational learning'.

The concept of 'organizational learning' first appeared in administrative language [16,17] in the late-1970s to mid-1980s. Today, this term has become increasingly powerful, and a great deal has been written about it in the literature. We could define it as: "the process by which entities, large or small, public or private, transform information into knowledge, disseminate and exploit it in order to increase their innovative and competitive capacity" [18] (p. 53).

Hitt [19] states that organizational learning is displayed by organizations with the skills to create, acquire, and transfer knowledge, as well as to modify their behavior to reflect the new actions and acquired knowledge. According to Castañeda and Pérez [20], organizational learning is understood as the process by which organizations, through individuals, acquire and create knowledge, for the

purpose of turning it into institutional knowledge, in such a way that it allows the organization to adapt to the changing conditions of its surrounding environment or to transform it, depending on its level of development.

Organizational learning is a dynamic process. According to Crossan, Lane, and White [21], it occurs over time and on three levels: Individual, group, and organizational (Figure 1). At each level there are subprocesses, which the authors refer to as the "4I" model: Intuition, Interpretation, Integration, and Institutionalization. They also signal tension between the assimilation of new knowledge (feedforward, from the individual to the organizational level) and the exploitation of what has already been learned (feedback, from the organizational to the individual level).

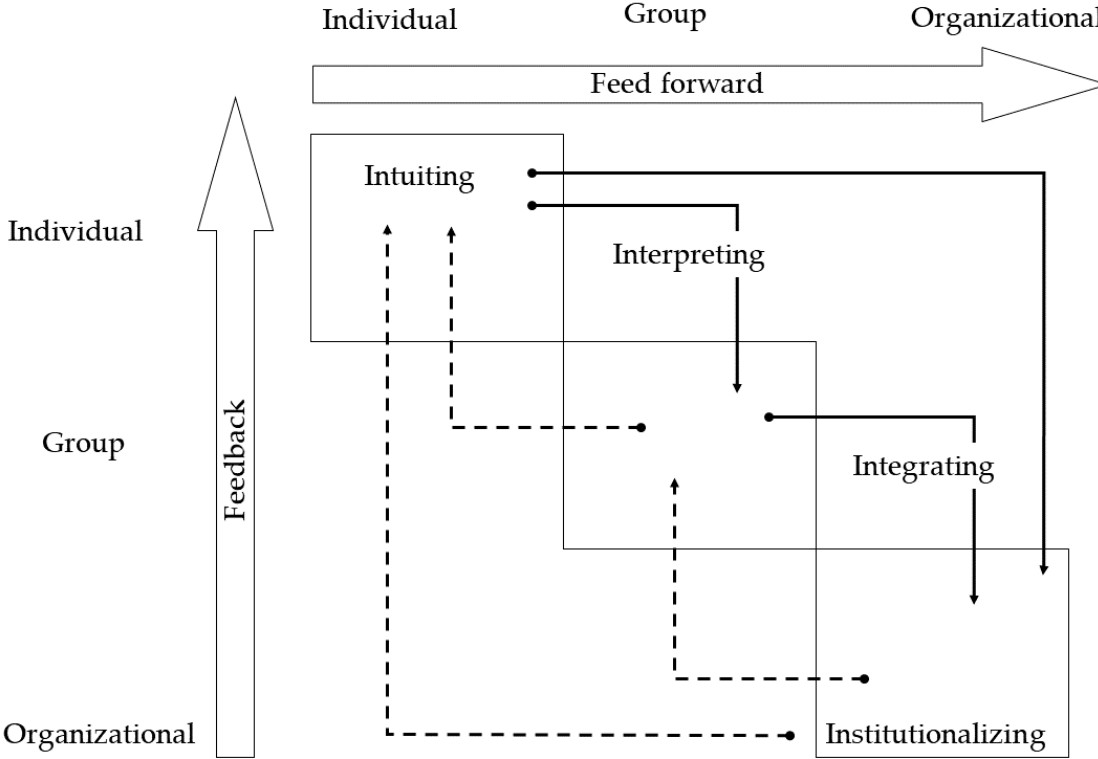

**Figure 1.** Organizational learning model. Source: Authors' own based on Crossan, Lane, and White [21] (p. 532).

The extensive literature consulted by Pérez and Cortés [22] also identifies those three levels at which learning takes place within an organization: Individual, group, and organizational. These authors explain it very clearly:

> "at an individual level, learning is affected by values, attitudes, personality traits, emotions, individual decision-making, perception, ethics, and motivation. At a group or team level, learning is studied in teamwork and behaviour, leadership, group decision-making, communication, power and politics, conflict, and negotiation. At an organizational level, learning is affected by organizational decision-making processes, organizational design, the role of technology, culture, and change". [22] (pp. 255–256)

Some authors refer to a fourth inter-organizational level [23], which could be defined as the learning that occurs through relations between organizations within the same corporate environment or guild.

Swieringa and Wierdsma [24], following the work of Argyris and Schön [16], presented a model (Figure 2) of organizational learning that allows us to differentiate between individual and collective learning, distinguishing three loops: Single-loop learning, which affects organizational rules and

norms (doing things better); double-loop learning, which refers to insights and vision (doing things differently); and triple-loop learning, which has to do with principles (doing other things) [25].

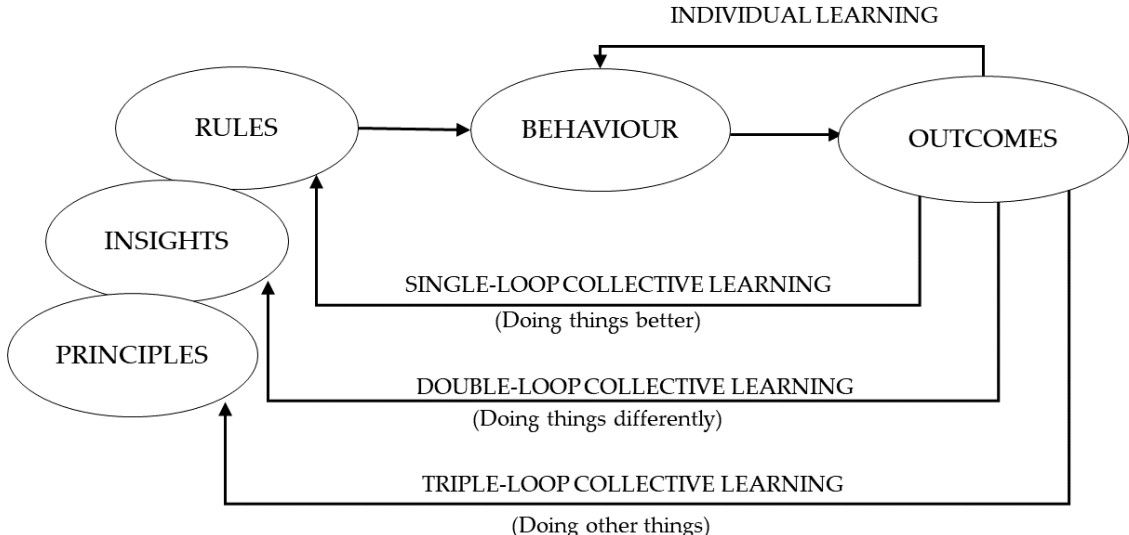

**Figure 2.** Individual learning and organizational learning. Source: Authors' own based on Swieringa and Wierdsma [24] and González, Zurriaga, and Martínez [25].

According to González, Zurriaga, and Martínez [25], interpreting Swieringa and Wierdsma [24], when an organization's norms lead to behaviors that carry positive expected outcomes, the objective is an individual learning of these norms and the continuance of organizational functioning. If the results are not as expected, the learning needed to change organizational norms requires collective learning, called 'single-loop' learning, which involves doing things better by changing the norms and practices of normal operation, thereby optimizing without making significant changes to the strategy, structure, culture, or systems.

If there are signs outside the organization that rule adjustment by itself is no longer adequate, there may be a need for new action strategies that change the set of strategic beliefs and shift the organization's position with regard to its surrounding environment. This raises the need to do things differently (double-loop learning), renewing the insights that will in turn modify the rules of the organization in order to maintain coherence.

Sometimes, the situation requires the organization to question its values, the essential principles on which it is based, the position it seeks to maintain, its own mission. According to the model summarized here, we would find ourselves needing to do other things (triple-loop learning), in other words develop new principles that, consequently, will also change insights and organizational rules.

Leadership, collaboration, and communication are part of the foundation for organizational learning [26]. Circular structures, organizational team dialogue, and fluid communication within the structure are factors that stimulate organizational learning [27]. A learning-oriented organization is one that facilitates the ongoing training and development of all its members and experiences, in itself, a continuous transformation [23]. Against the backdrop of the sustainability crisis defined at the start of this article, organizations need to be capable of learning and transforming in order to survive this context and also offer solutions to the challenges we are facing. The framework provided by the United Nations 2030 Agenda for Sustainable Development is even more urgent if we wish to achieve this transformation.

Although these models may not yet be mature enough to adequately reflect the underlying complexity, they appear to have introduced innovations that provide organizations with "the foundations for structuring the processes of cultural transformation destined to permeate the different areas of improvement, which not only support their sustainability, growth, and competitiveness, but also, as an

ethical foundation, fulfil their social responsibility of meeting the development and holistic wellbeing needs and expectations that society hopes to receive, as the principle that gives rise to each and every organization" [15] (p. 140).

According to Henríquez, Vallaeys, and Garzón [28], for an organization to be consistent with the principles of Social Responsibility, these should ideally permeate all organizational communications, culture, and decision-making. We believe that this transversality could also be expected within an organization that strives to be consistent with environmental sustainability. These authors highlight the need for a close relationship between Social Responsibility and organizational learning, as this process allows an organization to learn, change, innovate, and adapt to the surrounding environment, establishing shared goals, a culture. Such a relationship could be deemed to be essential between organizational learning and environmental sustainability, a fundamental part of any organization's Social Responsibility.

### 1.2. Universities, Sustainability Scenarios

From an organizational perspective, universities must closely monitor their surrounding environment to continuously adapt to its demands, proposing knowledge-based transformations as the main organizational resource for achieving social progress and development [29].

As noted previously, the 2030 Agenda, with its 17 Sustainable Development Goals (SDGs), is a guide to transformation that takes account of the most pressing global challenges. Because of their work generating and disseminating knowledge, and their position within society, universities are called upon to play a key role in achieving the SDGs. Indeed, it is unlikely that any of these goals can be met without the involvement of the university sector [30].

Building on that role, universities play a particularly important part in providing answers to the problems and challenges facing society now, and in the future [31]. They are also particularly important drivers of change for sustainability, since they train and educate future professionals who will have a direct or indirect influence on their environment [32], through their knowledge, values, and attitudes [33]. The organizational transformation of universities in the context of the sustainability crisis described above would appear to be an extraordinarily complex challenge.

However, work began in this direction many years ago. In recent decades, universities have moved on from the creation of 'green' campuses [34], to the pursuit of sustainability for university institutions as a whole [35]. In other words, the objective has evolved from establishing measures designed to manage activities in an environmentally responsible way, to developing strategies that will push them to become systemically more sustainable organizations, where environmental management is one of the elements to consider [8]. However, although they theoretically talk of sustainability, in practice, universities tend to focus just on the environment [36].

With major efforts, universities around the world are including environmental sustainability in different areas, such as management, research, or extension, with different levels of depth and success [37]. From the bibliographic review carried out, it could be concluded that, although both superficial and deep changes are being made, there seem to be hardly any models developed both to explain and to promote the integral sustainability process of universities, much less from the perspective of organizational learning.

Efforts to advance the sustainability of universities are often analyzed by differentiating between 'top-down' and 'bottom-up' strategies. Brinkhurst et al. [38] claim that both 'top-down' (generated within university governing bodies) and 'bottom-up' activities (mainly initiatives developed by the student body) have important strengths, but they believe that activities that emerge from intermediate positions, through staff members (faculty and administrative/services) and structures such as faculties and departments, have the greatest power to achieve long-term changes on the road to sustainability. They emphasize that the key strategies for empowering and motivating faculty members and staff are official permission and encouragement, and the development and institutionalization of staff involvement in decision-making.

For Conceição et al. [39], universities must offer their community the experience of a sustainable ecosystem grounded in the following principles: Connection with the surrounding environment and community, cooperation, closed-loop material cycles, and more thermodynamically efficient energy flows. The educational aspect should be added to this. The greatest challenge for the sustainability of universities lies within teaching since it implies a profound transformation [40], to effect the necessary global changes in the teaching-learning process [41]. Although higher education institutions are increasingly concerned about developing educational processes that will contribute to a more sustainable future, there is evidence that they do not yet understand the true nature of the changes required [42].

Most universities seem to remain focused on transforming environmental management, which can be an opportunity if properly combined with environmental education. The definition of this concept, agreed at the UNESCO-UNEP International Congress on Environmental Education and Training [43], emphasizes three core ideas: (1) The necessary acquisition or strengthening of environmental awareness, accompanied by (2) the provision of learning tools that (3) facilitate training and education for pro-environmental action.

As we can see, environmental awareness is the initial and fundamental pillar of effective environmental education. It can be defined as the system of experiences and knowledge that an individual actively uses in his or her relationship with the environment [44]. It is, therefore, a multidimensional concept, which encompasses interrelated knowledge, beliefs, values, attitudes, and behaviors linked to the environment. The applied learning of these facets induces motivation and competence, elements that De Castro [45] points to as prerequisites and determinants of effective pro-environmental behaviors.

Environmental education is clearly a permanent process toward competence for action [46], by incorporating the environmental variable into everyday decision-making in different areas of our life (personal, working, and social). It offers a useful tool for involving individuals and groups and making them responsible for the resolution of environmental problems, present and future. Environmental management and education are, therefore, closely related, and their coordination is key to achieving increased environmental awareness among members of the university community [33], as well as behavioral changes that help solve environmental problems [47,48]. This combination could provide the university community with that experience of a sustainable ecosystem mentioned previously, building on a perceived norm that respects the environment.

This concept, the perceived norm, was included in classic theories of social psychology about planned action devised by Ajzen [49] and has subsequently been used in countless studies in social and environmental psychology. This author notes that perceived behavioral control is determined by external and internal variables. The combination of attitude, subjective norm, and perceived behavioral control would result in behavioral intention [50], the most immediate precursor of behavior.

Many authors agree that universities require organizational changes to meet the challenges ahead. For example, one of the main challenges facing universities regarding their contribution to sustainable development is the need to tackle complexity and introduce systemic thinking by overcoming the problem of disciplinary division [51–55]. Leff [6] states that the division of learning into disciplines has been institutionalized in universities, generating interests linked to dominant academic practices and blocking the transformation of these institutions and the renewal of existing curricular structures and contents.

These and other areas of resistance displayed by higher education institutions in making substantial changes to their structures mean that sustainability proposals hit a 'glass ceiling', which highlights a considerable gap between the sustainability goals formulated in declarations and strategies and the achievements ultimately attained through concrete measures [56].

*1.3. Case Study of the University of Córdoba (Spain)*

Over the past two decades, the University of Córdoba (Spain), hereinafter UCO, has been gradually incorporating the environment into its policies, structures, and action plans. Like other universities, it has also encountered significant limitations and barriers. However, progress overall appears to have been quite positive. This can be deduced from the application of the "self-diagnostic tool for environmental sustainability in Spanish universities" [57,58], an extremely useful instrument developed by the CRUE Sustainability Commission in 2010. It is tailored to Spain's university system and oriented toward self-diagnosis and continuous improvement of university environmental sustainability. Currently, the tool incorporates 133 progress indicators grouped into three areas (organization, teaching and research, and environmental management).

UCO has been using this tool as a reference and roadmap for planning environmental action since its very first edition. In the most recent self-assessment for 2019, UCO scored 67.5%. A simulation of the score the University would have obtained 20 years ago, in the year 2000, yielded a value of 7.1%, thus indicating positive progress during this time.

In this article, we will seek to explain the main actions carried out over these past 20 years, the stages we can detect in the process, and which variables influence individual, group, and organizational levels. Is it possible to propose an organizational model within the framework of complexity that integrates the perspective of organizational learning with aspects such as environmental awareness or the perceived norm? What would the key elements of such a model be to optimize an organization's learning flows and thus accelerate the environmental sustainability process within universities? Our main hypothesis focuses on that the combined use of environmental awareness and the perceived norm can enhance the perspectives of organizational learning, optimizing the understanding of the sustainability processes at universities.

We hope to provide some useful reflections on universities as organizations that need to learn within a crisis of sustainability, taking into account their complexity with a view to evolving and offering solutions to society.

## 2. Materials and Methods

The aim of this descriptive-explanatory study is to identify possible variables present in environmental sustainability processes within universities in the context of organizational learning, so as to characterize the actions and strategies undertaken. The design is transversal, studying this process in the case of UCO over a period of 20 years, between 2000 and 2020.

Firstly, a literature review was conducted, focusing on development and organizational learning models. The search engines and databases of scientific journals such as Google Academia, Web of Science, Scopus, SciELO, Redalyc, and RECYT were used, selecting the following criteria and variables based on their alignment with the subject matter examined:

- In accordance with the model devised by Crossan, Lane, and White [21], we have used the concepts of feedforward (from individual to organizational) and feedback (from organizational to individual) to characterize the direction in which the action of learning starts and flows.
- Following the model developed by Swieringa and Wierdsma [24], we felt it would be interesting to use the concept of collective learning loops, "doing things better", "doing things differently", and "doing other things".
- In addition, based on the analysis carried out in the theoretical framework, the authors stressed the need to incorporate criteria referring not only to the direction of the action, but also to the type of intended effect or outcome: Those seeking to strengthen the environmental awareness of the university community and those seeking to act on the perceived norm, that is, on the internal and external perception of a university increasingly committed to sustainability.

In this line of analysis, using the criteria defined previously, the following categories and subcategories were established with the aim of characterizing the actions undertaken:

- Direction of the measure (D):

  o D1: Top-down: Feedback, working on the perceived norm, considering here measures that emanate from the University's governing bodies.
  o D2: Bottom-up: Feedforward, working on environmental awareness, including here measures that attempt to filter up from the community to governing bodies, and those pertaining to technical and scientific structures, or centers that design, coordinate, execute, and redistribute, mainly horizontally within the organization.

- The level at which the measure is anchored (L):

  o L1: Group.
  o L2: Organizational.
  o L3: Inter-organizational.

- Objective of the measure (O):

  o O1: Do things better, influence internal rules.
  o O2: Do things differently, influence insights.
  o O3: Do other things, influence the principles of the organization.

The actions and strategies carried out at the University of Córdoba in the field of environmental sustainability over the past 20 years were then identified. To this end, the university's environmental management reports for each year were analyzed [59]. An initial comprehensive inventory was made of actions taken, in chronological order and by area of action, selecting the events considered to be the most relevant in the period of study in the field of environmental sustainability. The aim was to operationalize the model and analyze the milestones that would have generated decisive transformations or advances. These actions were characterized according to the criteria described above.

Finally, the analysis of the information obtained through the characterization of actions according to the variables discussed was used to identify the different stages of progress in the process, and to propose an organizational model capable of ordering variables into a system within the framework of complexity.

## 3. Results

Reviewing UCO's environmental management reports between 2000 and 2020, a total of 90 actions were identified in the following areas: Institutional policy, education and participation, indicators, energy, water, biodiversity, waste, procurement, and mobility. At the same time, 70 scientific contributions related to the actions undertaken were also recorded (papers delivered at national and international congresses, scientific publications, books and book chapters, doctoral theses, undergraduate and master's degree dissertations). From that initial list, the 15 most relevant ones were selected, those that have served as a basis or reference for subsequent actions or to modify the scenario, leaving aside the actions or activities that have not transcended the specific intended effects or outcomes (Table 2).

**Table 2.** Characterization of the actions analyzed. Source: Authors' own.

| Action No. | Name of Action | Direction of the Measure (D) | Level at Which the Measure Is Anchored (L) | Objective of the Measure (O) |
|---|---|---|---|---|
| 1 | Create a specific structure for environmental management and provide the required human and economic resources ('Environmental Protection Service-SEPA') | D1 | L2 | O3 |
| 2 | Sign up to the National Network on Sustainability at University Level ('CRUE Sustainability Commission') | D1 | L3 | O3 |
| 3 | Conduct an initial environmental diagnosis | D1 | L2 | O3 |
| 4 | Institutional Environmental Policy Statement | D1 | L2 | O3 |
| 5 | Government subsidies for environmental improvement | D2 | L1 | O3 |
| 6 | Partnership between SEPA and the master's degree in Environmental Education | D2 | L1 | O1 |
| 7 | Sign up to the Regional Network on Environmental Education and Participation ('Andalucía Ecocampus Programme' run by the Andalusian Regional Government), and create the structure and resources to coordinate it ('Sustainability Office') | D1 | L2 | O3 |
| 8 | Partnership between SEPA, Cooperation Area, and Procurement Service | D2 | L1 | O1 |
| 9 | Implementation of quality and environmental management system in SEPA | D2 | L1 | O1 |
| 10 | Diagnosis and Participatory Environmental Action Plan 2013 | D2 | L2 | O2 |
| 11 | Create an in-house system of certification for good environmental practices ('Clover Programme') | D2 | L1 | O2 |
| 12 | Incorporate SEPA into the University Infrastructure Area | D1 | L1 | O1 |
| 13 | Partnership between SEPA and the Directorate-General for Energy Efficiency | D2 | L1 | O1 |
| 14 | Public commitment to social responsibility and sustainability | D1 | L2 | O3 |
| 15 | I Environmental Sustainability Plan 2019–2022 | D2 | L2 | O3 |

Note: D1: Top-down: Feedback, perceived norm; D2: Bottom-up: Feedforward, environmental awareness, L1: Group; L2: Organizational; L3: Inter-organizational; O1: Do things better; O2: Do things differently; O3: Do other things.

This set of actions was characterized according to the criteria described above (Table 3).

**Table 3.** Results of the actions characterized. Source: Authors' own.

| D | O1 | O2 | O3 | L |
|---|---|---|---|---|
| | 12 | - | - | L1 |
| D1 | - | - | 1, 3, 4, 7, 14 | L2 |
| | - | - | 2 | L3 |
| | 6, 8, 9, 13 | 11 | 5 | L1 |
| D2 | - | 10 | 15 | L2 |
| | - | - | - | L3 |

Analysis of Table 3 shows that the actions are mainly concentrated in two profiles:

- Those characterized as D2, O1, L1 (Actions 6, 8, 9, and 13), that is, launched at a group level with a view to 'doing things better' and in the bottom-up direction towards a strengthening of environmental awareness. These are mainly partnerships between internal agents generated from intermediate positions within the university community, which influence the internal rules or regulations of the organization.
- Those characterized as D1, O3, L2 (Actions 1, 3, 4, 5, and 14), developed at an institutional level with a view to 'doing other things' and in the top-down direction towards a strengthening of the perceived norm. These are strategic commitments and decisions arising from within the University's governing bodies, affecting organizational principles, or incorporating new structures or instruments that ultimately have an important influence on the whole organization and on each of its members.

It should also be noted that, of all the actions analyzed, only two of them are in the O2 category ('doing things differently'): Actions 10 (Participatory Environmental Diagnosis and Action Plan) and 11 (Create an in-house system for certification of good environmental practices in UCO's 'Clover Programme'). These are two eminently participatory measures, which required a great deal of organization and mobilization initially to slot into the dynamics of the University, but which were hugely successful, ushering in major transformations and changes in direction.

The identification and characterization of the actions according to the proposed criteria has enabled us to determine important chronological milestones in the process followed by UCO:

- Milestone 1 (2001). Creation of a specific structure for environmental management and allocation of the required human and economic resources.
- Milestone 2 (2005). Public institutional commitment to ongoing and integral environmental action and improvement of the organization. Strengthening of human resources for the environmental management structure. Positioning universities as key environmental education settings.
- Milestone 3 (2009). Partnership between technical environmental management unit and the academic area of environmental education: configuration as a scientific-technical structure.
- Milestone 4 (2012). Consolidation of the scientific-technical structure as a reference point in university environmental management and education. Strengthening of Human Resources.
- Milestone 5 (2018). Incorporation of environmental sustainability as a 'core value' and strategic area by the institution.

In turn, these milestones make it easier to spot the different stages of progress within the process, focusing on the characteristics of each of them (Figure 3).

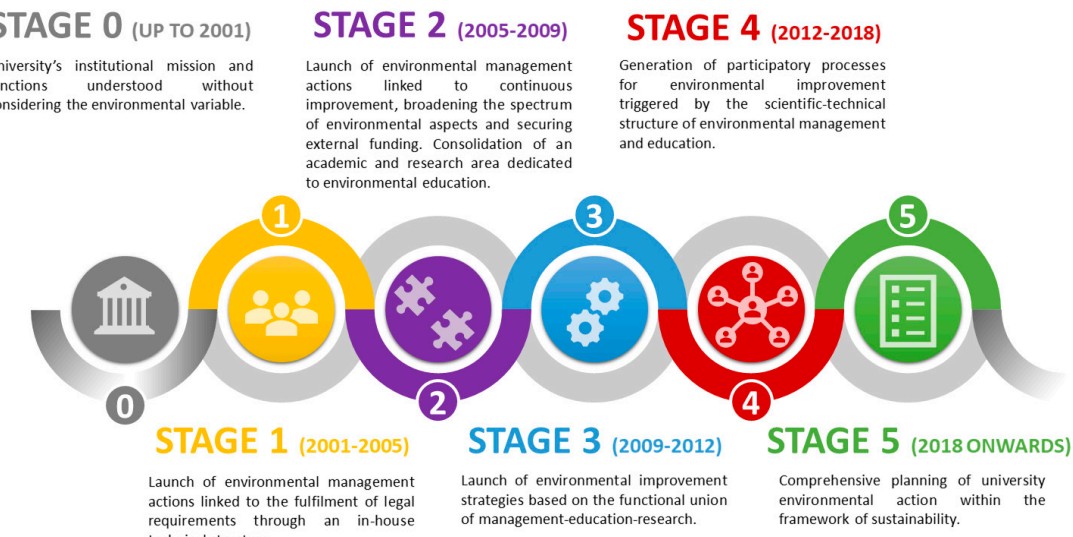

**Figure 3.** Stages detected in the process to achieve environmental sustainability followed by University of Córdoba (UCO) over the last 20 years. Source: Authors' own.

Having identified possible variables to characterize the actions and processes present in the pursuit of sustainability within universities in the context of organizational learning, the next step is to define a proposed organizational model that can order these variables into a system within the framework of complexity (Figure 4).

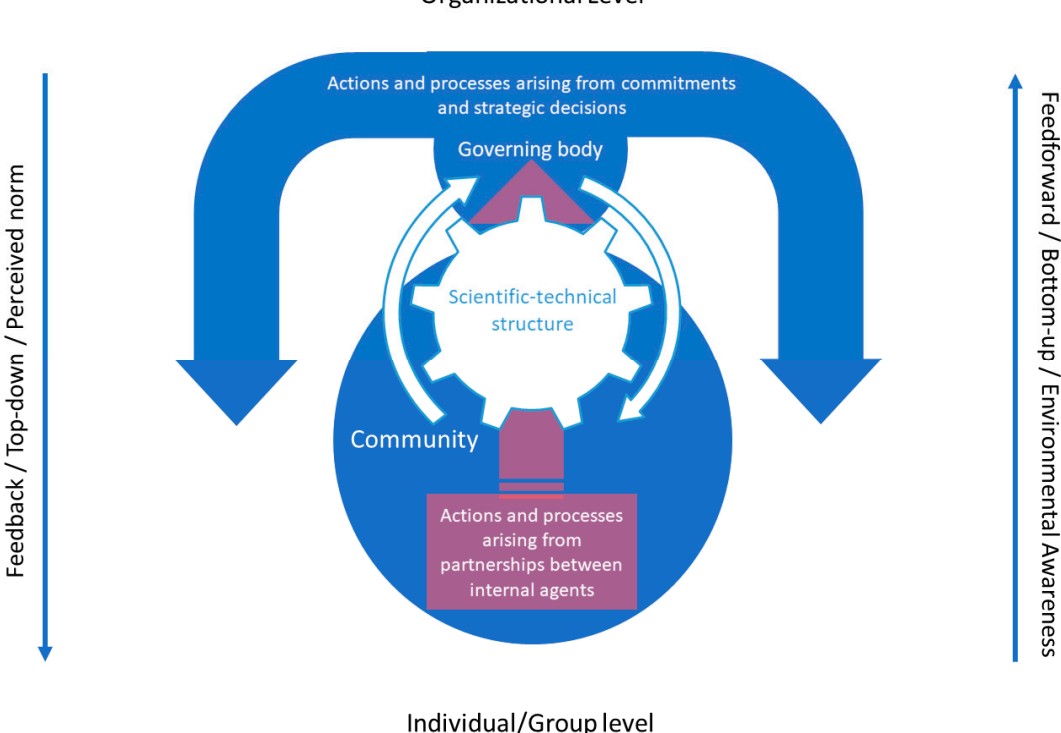

**Figure 4.** Proposed organizational model for a university that is learning environmental sustainability. Source: Authors' own.

The model incorporates criteria used to characterize actions and processes within the framework of organizational learning, such as the direction and the level from which they are initiated. It also identifies the main actors: Governing body, scientific-technical structure, community, and internal

partners (although external partners such as inter- or supra-organizational entities do not appear explicitly, they can also be considered at a higher level). Actions and processes flow in two directions that feed back into one another: On the one hand, within the university community, actions and processes emerge, arising mainly from partnerships between internal agents, with a bottom-up approach from the individual/group level to the organizational level. This feedforward direction makes it possible to strengthen environmental awareness at all scales. On the other hand, in a complementary and synergistic way, the governing body reacts by developing through feedback a series of actions and processes arising from strategic commitments and decisions, which permeate all structures and collectives in a top-down direction, acting on the perceived norm. This model highlights the role of a scientific-technical structure as a stabilizing force, a catalyst, facilitator, attractor, and to a large extent executor of transformative flows within the organization, which could be a key factor in its evolution towards sustainability. These flows, taking into account the context of complex organizations such as universities, could push the system towards higher maturity levels, thus generating a spiral of continuous improvement in the organization's environmental performance, associated with progressive levels of environmental awareness and the perceived norm.

## 4. Discussion

Many institutional decisions are urgently needed to address the profound changes required to transform the current socio-economic system into a much more sustainable one [60]. Education is undoubtedly one of those fundamental pillars that must be aligned with this idea of transformation. In particular, universities must take a leading role in the necessary transition of societies in order to play their fundamental role in them [31,32].

To become true social leaders in the transition to a genuinely sustainable world, universities need a global institutional strategy [61] that is well coordinated and adequately supported at the supra and inter-university levels. Universities should view themselves as places of learning and experience for sustainable development and should, therefore, focus all their processes on the principles of sustainability [62].

The framework provided by the United Nations 2030 Agenda could accelerate these changes. Across all their areas, universities already make significant contributions to the achievement of the SDGs. However, for the SDGs to be achieved globally, universities must become advocates for sustainable development and play a leading role in implementing these goals. This poses some major challenges, but the 2030 Agenda can provide a perspective and an incentive to work on structural and organizational solutions that help accelerate the contribution made to local, national, and global wellbeing [30].

The path to achieving sustainable societies, as well as universities, requires multi-factorial approaches and models that incorporate complexity. Along this path, we need to study examples of positive progress, which can be extrapolated to other instances, and which must be based on the experimentation of theoretical models consistent with complexity.

In this new paradigmatic approach, small actions can have unpredictable consequences, implying a different approach to organizational management [14,15], within universities as well. Understanding universities as organizations with the ability to experience collective learning, not just individual learning, would seem to be a coherent way of exploring answers that, within the framework of complexity, will help to turn universities into sustainable ecosystems. Thus, the promotion of aspects that facilitate 'organizational learning' opens the door to innovation within universities.

The experience gained throughout these years at UCO has provided information and knowledge on how we could generate greater environmental awareness at the University of Córdoba, which leads to a more sustainable institution and better integration of sustainability into the skill sets of graduates. To achieve this, we have accessed classical sources of environmental education and psychology and have been testing a variety of strategies, actions, and research, progressively shaping a model of action for this university.

Two ostensibly complementary perspectives have been a lynchpin in the evolution of UCO's pursuit of environmental sustainability and in the preparation of this study.

The first is environmental awareness: The theoretical approach taken to this important variable of pro-environmental behavior is through the identification of different facets consisting of complex interrelated psychological processes, such as knowledge, beliefs, values, and attitudes as triggers of behavioral intention and manifest pro-environmental behavior. This multidimensional conception (proposed by Chuliá [63], around four dimensions: Cognitive, affective, conative, and active) feeds off a reinforcing relationship between dimensions that can trigger progressive and continuous spirals toward increasingly mature states of environmental awareness.

Any environmental education strategy should include among its main objectives the achievement of significant gains in the environmental awareness of the recipients [33]. The activation and strengthening of environmental awareness within the university community can be configured as a driver and guide, generating an active variable in decision-making at all scales, all levels of responsibility, and within all collectives. The proposed model shows a continuous flow of community environmental awareness activation that, in turn, generates the need for increased commitment within governing bodies to further strengthen awareness and to make visible an increasingly rigorous level of perceived norm.

This latter element, perceived norm [49,50], is the second perspective to which the model refers. The environmental sustainability of universities is constructed not only as the sum total of the environmentally sustainable attitudes of each and every member of the university community. In addition to this, the whole organization must also condition the pro-environmental behavior of university students. The study carried out by Gomera, Villamandos, and Vaquero [33] indicates that the environmental awareness of students at UCO had not been influenced throughout their undergraduate studies (this research opened the door to a series of studies aimed at measuring this indicator in different contexts, presented in Villamandos, Gomera, and Antúnez, [64]). This was an important finding to step up efforts in the combination of actions aimed at influencing environmental awareness through the perceived norm.

If the system is perceived by individualities as an environmentally sustainable 'whole', it will have a clear impact on the way in which individuals behave and, therefore, on the shaping of their own environmental awareness. As a result, the system as a whole would be strengthened towards ever higher levels of environmental sustainability. Of course, these kinds of issues are much easier to discuss than implement effectively. The strategy pursued over these years at UCO has sought to put different actions into practice in order to achieve this.

It is therefore based on a series of theoretical references, taken as axioms, to support this model:

- Significant achievements cannot be made in a social context without increasing the environmental awareness of its members.
- The sustainability of a social system is based on the behavioral models of its members.
- Human behaviors are not rational and are influenced more by environmental circumstances than by knowledge.

Sustainability variables and references also change over time. On the one hand, because the overall situation and the perception of gravity and intensity vary over time. On the other hand, because, as we move forward on the path to achieving sustainability at a given university, new challenges and goals gradually emerge. Consequently, and to paraphrase the documents of the International Union for the Conservation of Nature (IUCN), in conservation education, "the system must be constantly monitored" to lead us to a state of permanent adaptation [65].

UCO has been working in this regard for more than 20 years. After pondering which variables might play a more prominent role in order to influence them, eventually, we decided to pursue a model that would integrate an effective and functional combination of environmental management and education to strengthen both environmental awareness and the perceived norm. These two

variables must drive and guide the process, generating an active basis for decision-making at all scales and in all areas (governing bodies, university areas related to or separate from sustainability, faculty, administration and services staff, students, etc.).

The analysis conducted of the main environmental actions carried out at UCO over the last 20 years indicates that there are two main groups: ctions undertaken at a group level with a view to 'doing things better' and in a bottom-up direction towards strengthening environmental awareness; and actions developed at an institutional level with a view to 'doing other things' and in the top-down direction towards a strengthening of the perceived norm. This differentiation demonstrates that bottom-up actions only achieve organizational learning that affects internal rules (doing things better), but top-down measures have had the most important influence by affecting institutional principles or incorporating new parts into the organization (doing other things).

On the other hand, the analysis identifies only two actions related to 'doing things differently': Two participatory measures that, although they might appear to be small, have had a significant impact on the evolution of the system.

The first of these was the Participatory Environmental Diagnosis and Action Plan, which was planned and executed through an integral and complex process of participation, involving the university community in improving the environmental management of the institution, whilst at the same time, efforts were being made to create environmental awareness and improve environmental education for participants. This experience showcases an approach that emphasizes the importance of participatory methodologies in the search for lasting changes towards more sustainable practices in universities [66].

The second of the measures was the creation of an in-house system for the certification of good environmental practices within UCO, called the 'Clover Programme' [67]. UCO's Clover Programme provides a useful and interesting tool for the achievement of the 2030 Agenda within universities, both for the education and training provided and as a mechanism that helps universities to advance through the examples set by their own members [68]. This program is the most practical embodiment of the proposal put forward in this article to combine environmental management and education linked to transformative action as a fundamental component.

In addition, these examples of participatory actions with a significant impact on the organization tie in with factors that stimulate organizational learning, such as circular structures, organizational team dialogue, fluid communication, collaboration, and lifelong learning [23–27].

The model provided in this study, although based on the experience of UCO, is intended to be a reference for other higher education institutions, which we invite to compare with their own evolution and dynamics.

Likewise, we can find certain parallels between the proposed model and the approaches of the Science of Complexity described by authors such as Morin [11], contexts of uncertainty [69], or Prigogine's Chaos Theory [9], and their usefulness in managing complex systems [15].

Thus, Prigogine's idea of thermodynamically open systems subjected to a constant flow and dissipation of energy illustrates how organizational systems, such as universities, become more complex and mature as they are subjected to pressure that seeks to set them on the path to sustainability. This approach is consistent with bottom-up strategies and justifies how the pressure that gradually increases environmental awareness among university students has the capacity to take the system as a whole to a progressively more sustainable and qualitatively different level.

Morin's holographic principle, according to which the whole is inscribed in some way in each of the parts, shows how 'the whole' also determines the parts and not just the other way around. Thus, if the whole is, or is perceived to be, a sustainable entity, it will condition the constituent elements. In our case, the whole, the university, is perceived as a highly sustainable institution, one that is committed to sustainability, and this perception will determine the behavior of its members, somewhat independently of their level of environmental awareness. For this reason, the process must be managed with an explicit focus on the university's perceived image both internally and externally. Moreover, one of the university's priorities should be to maintain and permanently improve that image. Clearly,

this approach is in complete alignment with the idea of 'perceived norm' that comes from the field of Social Psychology, as well as the definition of 'top-down' strategies described above.

Looking at the model from the perspective of Prigogine's Chaos Theory [9], we can see a complex system in a state of 'non-equilibrium'. This situation is enhanced by a continuous influx of energy for environmental action and awareness. This continuous influx might have placed the system at a so-called point of bifurcation [70], that unique moment in dissipative and complex systems in which the direction the system will take in a crisis is not predictable. At UCO, that unique moment is approaching and, obviously, it is a critical state in which the most active and conscientious agents with regard to sustainability must continue to draw pro-environmental energy into the system.

The example of the organizational evolution experienced by UCO in relation to environmental sustainability, visible in the different stages identified and characterized, set out in the results section, allows us to hypothesize three major stages in the pursuit of sustainability within universities (Figure 5).

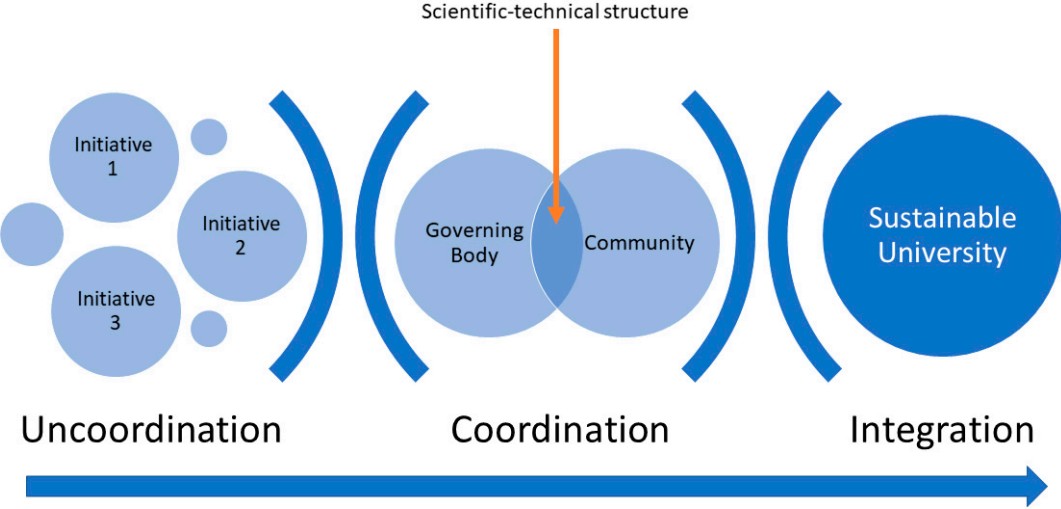

**Figure 5.** Three major stages in the evolution of a learning university on its path to sustainability. Source: Authors' own.

The first stage is one of 'uncoordination', in which initiatives are beginning to emerge and be implemented, mainly to comply with legal provisions, and in groups that are largely unconnected. One of these groups could be the scientific-technical structure in its initial form.

The second stage is one of 'coordination', during which the different groups coordinate by establishing partnerships, the organization clarifies its environmental mission and vision and strengthens the role of the scientific-technical structure as a key component that helps harmonize and optimize environmental actions.

The final stage would be one of 'integration', in which a dynamic characterized by a robust environmental organizational culture would be achieved, as an unconscious competence [46], that structures all levels and areas of the university and establishes processes of continuous environmental improvement within the framework of sustainability, perceived as the norm and developed through integral planning.

Returning to the experience of UCO, the first Environmental Sustainability Plan could be viewed as a promising advance that is pushing UCO towards the 'integration' phase. To see whether this is achieved, it will take years to study whether the organization has attained a higher level of complexity. This plan, developed by means of a participatory process, includes the variables studied in recent years and part of the results of the management actions undertaken.

The sustainability of universities is neither a simple nor a static objective. It is actually a road, designed to be travelled together for many years. Today's achievements are the basis of tomorrow's proposals, but not just that. The objectives and emergencies of sustainability also change; so, the goals

cannot remain immutable. The road must be constantly redesigned and permanently monitored. Not only to see how the system as a whole reacts to the actions we take, but also to make sure we have not strayed from the global current. A strategic trend that is constantly redefining the fight for sustainability and which demands that we keep in step with this global strategy. This constant redefinition has nothing to do with improvisation, but with incorporating the need for constant readjustment into the evolution of complex systems. A challenge we have sought to shed some light on by means of the analysis carried out, the proposed model, and the reflections derived from it.

## 5. Conclusions

We believe that the model we have presented introduces several relevant ideas, which we detail as conclusions:

- The scientific and technical structures responsible for the environmental management of universities could be the key component of the system if they succeed in acting as attractors, catalysts, or facilitators of environmental disturbance flows in the system. A strategic perspective that links environmental management closely with environmental education is essential to this. Moving towards acting as a single functional scientific-technical unit with a research vocation linked to action would appear to bring us a step closer towards optimizing that key component.
- Top-down and bottom-up measures are complementary and, if well-coordinated, can result in a positive feedback loop in organizational learning about environmental sustainability at university. The two indicators identified here, in relation to which action has been taken and which provide a better understanding of this loop, are the perceived norm and environmental awareness. Measures that act on the environmental awareness of individual members of the university community generate a more favorable environment for top-down organizational changes that end up re-adapting the perceived norm, drawing it closer towards the notion of a 'sustainable ecosystem'. This evolution of the perceived norm in turn triggers a need to increase the measures and their ambition in order to further increase the environmental awareness of individuals. Logically, there can be limitations to this loop, linked to the context of each university or of each period in time. These might include a governing body that blocks the measures necessary to materialize organizational learning consistent with the level of environmental awareness of its university community, or the absence or weakness of a scientific-technical structure to induce the flows of the loop.
- Measures commonly referred to as bottom-up have been identified as those related to actions and processes arising from partnerships between internal agents. This characterizes flows that, in different ways (some planned and others not), eventually rise up and push the organization towards more complex levels of environmental performance. There are also partnerships between internal agents that generate horizontal flows, which feed back into the system and 'accumulate energy' until they too can rise up and feed into this loop.
- An organizational learning approach [21,24] would appear to offer a better understanding of the flows generated within a university. Understanding universities as organizations capable of learning could contribute to thinking about ways of doing things in a manner consistent with the educational and scientific spirit of these institutions and with the necessary transformation towards 'sustainable ecosystems'.

**Author Contributions:** Conceptualization, all; introduction, M.A., methodology and results, A.G. and M.A., discussion, all; writing—review and editing, all; supervision, F.V. All authors have read and added to the published version of the manuscript.

**Funding:** This research received no external funding.

**Acknowledgments:** The authors of this paper would like to thank the research group to which we belong ('SEJ-049: Educational Evaluation and Innovation'), and especially its director, Ignacio González, for the support and advice offered throughout the process of preparing this work. We wish to express our special thanks to all past and present members of the Environmental Protection Service (SEPA) team (Manuel Vaquero, Ana de Toro, J. Emilio Aguilar, Clara Guijarro, Justo P. Castaño) and the Sustainability Office (Yolanda León, Bárbara Martínez), for their professionalism, dedication, and motivation to help the University of Córdoba move ever closer to the integration of environmental sustainability. And, of course, UCO's university community and its governing team, for their commitment to a more sustainable university.

**Conflicts of Interest:** The authors declare no conflict of interest.

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
