# Peer review of "Universities That Learn to Tackle the Challenges of Sustainability: Case Study of the University of Córdoba (Spain)"

_sustainability, doi:10.3390/su12166614_

Round 1

Reviewer 1 Report

The topic of this paper is very important and interesting. The paper analyzes the progress of the University of Córdoba (Spain) - UCO, in the past 20 years, when it comes to the process of involving universities in environmental protection. During the mentioned period, the UCO carried out various actions in the domain of sustainable development and thus went through several stages, which were later described in the paper. The authors analyzed this process and proposed a unique organizational model intended for higher education institutions in order to learn about the importance of including sustainable development in university activities.

This manuscript shows significant scientific background to provide a new model in this field. Nevertheless, I am of the opinion that a new chapter should be added in this section that provides an overview of the already existing models that have been applied at other universities in the world.

The English language is appropriate and understandable. However, the paper must be created in English only, so you are kindly asked to translate all references into English. Of course, you can keep the original references in brackets.

The article is well written and analyzes is presented appropriately. However, although some sections are not mandatory, it is recommended to be added to the manuscript because of complexity of the topic. From my point of view, I suggest making the names of chapters in the introduction section, and also putting the conclusion part (because discussion is unusually long).

The research question is original and the results are interpreted appropriately. Nevertheless, I am of the opinion that the hypotheses of the paper should be more clearly emphasized in the text itself.

Reviewer 2 Report

Tha manuscript is interesting and has practical value. Should be a good point to conduct similar researches at other universities in the European Union, or even outside the EU.

Comments and remarks:

  • A very good "Introduction" and a very good explanation of the researched subject. I would feel the need of some changes: all the parts that are very practical and go hand in hand with the "Materials and Methods" section - such as lines 77-84 and some other more - I would include them there, in the second section, as they are describing in detail the methods used to conduct your research. The "Introduction" should be about reviewing the current state of the research and cite the key publications mainly. 
  • In the "Introduction" I would present the need to conduct such research on universities earlier in the explanations, as they are the core of your article.  (the idea of "university" appears at line 153)
  • At line 47 - you are saying the 3 theories are equals (the Chaos Theory, the transdisciplinary and the perspective of complexity)? If not, please state why you have chosen to deal with the third one. If yes, please state from which point of view (for example, the complexity theory is a field of theoretical computer science and mathematics, while Chaos Theory is a field of mathematics). I would advise more precision here, at these concepts.
  • Sources needed for Tables 2 and 3
  • The "Discussions" part is too long (lines 380-590). I would advise adding a "Conclusions" section.
